# Comparative Analysis of Serum and Serum-Free Medium Cultured Mesenchymal Stromal Cells for Cartilage Repair

**DOI:** 10.3390/ijms251910627

**Published:** 2024-10-02

**Authors:** Meiqi Kang, Yanmeng Yang, Haifeng Zhang, Yuan Zhang, Yingnan Wu, Vinitha Denslin, Rashidah Binte Othman, Zheng Yang, Jongyoon Han

**Affiliations:** 1Critical Analytics for Manufacturing Personalised-Medicine (CAMP) Interdisciplinary Research Group (IRG), Singapore-MIT Alliance for Research and Technology (SMART) Centre, Singapore 138602, Singapore; meiqi.kang@smart.mit.edu (M.K.); yanmeng.yang@smart.mit.edu (Y.Y.); rashidah.othman@smart.mit.edu (R.B.O.); 2Department of Orthopaedic Surgery, Yong Loo Lin School of Medicine, National University of Singapore, Singapore 119288, Singapore; haifeng.zhang@shgh.cn (H.Z.); zhangyuan@cqmu.edu.cn (Y.Z.); doswuy@nus.edu.sg (Y.W.); isiadjd@nus.edu.sg (V.D.); 3NUS Tissue Engineering Program, Life Sciences Institute, National University of Singapore, Singapore 117510, Singapore; 4Department of Biological Engineering, Massachusetts Institute of Technology, 77 Massachusetts Avenue, Cambridge, MA 02139, USA; 5Department of Electrical Engineering and Computer Science, Massachusetts Institute of Technology, 77 Massachusetts Avenue, Cambridge, MA 02139, USA

**Keywords:** mesenchymal stromal cell, serum-free medium, osteochondral defect model, cartilage regeneration, chondrogenesis

## Abstract

Mesenchymal stromal cells (MSCs) are promising candidates for cartilage repair therapy due to their self-renewal, chondrogenic, and immunomodulatory capacities. It is widely recognized that a shift from fetal bovine serum (FBS)-containing medium toward a fully chemically defined serum-free (SF) medium would be necessary for clinical applications of MSCs to eliminate issues such as xeno-contamination and batch-to-batch variation. However, there is a notable gap in the literature regarding the evaluation of the chondrogenic ability of SF-expanded MSCs (SF-MSCs). In this study, we compared the in vivo regeneration effect of FBS-MSCs and SF-MSCs in a rat osteochondral defect model and found poor cartilage repair outcomes for SF-MSCs. Consequently, a comparative analysis of FBS-MSCs and SF-MSCs expanded using two SF media, MesenCult™-ACF (ACF), and Custom StemPro™ MSC SFM XenoFree (XF) was conducted in vitro. Our results show that SF-expanded MSCs constitute variations in morphology, surface markers, senescence status, differentiation capacity, and senescence/apoptosis status. Highly proliferative MSCs supported by SF medium do not always correlate to their chondrogenic and cartilage repair ability. Prior determination of the SF medium’s ability to support the chondrogenic ability of expanded MSCs is therefore crucial when choosing an SF medium to manufacture MSCs for clinical application in cartilage repair.

## 1. Introduction

Mesenchymal stromal cells (MSCs), characterized by their inherent self-renewal property [1], multipotent differentiation potential [2], profound immunomodulatory capacities [3], and availability from multiple adult and prenatal tissues, such as bone marrow [4], dental pulps [5], adipose tissues [6] and umbilical cord [7], have emerged as promising candidates for regenerative medicine applications. According to the International Society for Cellular Therapy (ISCT), the characterization of MSCs should at least satisfy the criterion of their plastic adherence under standard culture conditions; the ability to differentiate in vitro toward chondroblasts, adipocytes, and osteoblasts; the lack of hematopoietic markers; as well as the expression of surface molecules such as CD90, CD105, and CD73 [8]. The fact that MSCs are inducible to differentiate into progenitors of many mesodermal lineages, as well as ectodermal (neurocytes) [9] and endodermal lineages (hepatocytes) [10], unlocks the potential to address a wide array of unmet clinical needs. To date, there are more than 1300 registered clinical trials with MSCs as treatment options, out of which 30 are concerning the use of MSCs or their derivatives for cartilage repair.

Articular cartilage possesses minimal regenerative capacity and, therefore, necessitates effective restoration strategies for individuals experiencing joint trauma or osteoarthritis—a condition affecting more than 500 million of the global population with a significant socio-economic burden [11]. Numerous surgical interventions exist, including microfracture, arthroscopic debridement, perichondral and periosteal arthroplasty, autologous osteochondral transplantation, and autologous chondrocyte implantation (ACI) [12]. These methods, however, often fall short due to the inferior quality of the tissue repaired, graft morbidity, and wearing of replacement components. Evidence from preclinical and clinical trials suggests MSC transplantation has achieved clinical outcomes comparable to ACI, positioning MSCs as a promising treatment option for articular cartilage repair [13].

The therapeutic utilization of MSCs, or the application of MSC-derived products like exosomes, necessitates in vitro expansion to reach therapeutic dose, given the typically insufficient yield of cells upon initial isolation. Traditional MSC culturing protocols have relied on the incorporation of 10% fetal bovine serum (FBS), which facilitates cell attachment and provides essential nutrition and growth factors [14]. However, the use of FBS poses risks of xeno-contamination, virus, prion, bacterial infections, and batch-to-batch variations [15]. The consensus among industries is to endow an SF culture system aimed at reducing variability, enhancing traceability, and ultimately achieving a controlled pipeline that is critical for regulatory approval [16,17]. Additionally, the use of a fully chemically defined SF medium with the incorporation of various growth factors and bioactive compounds has benefits such as increase in cell proliferation and MSC population homogeneity, reduced donor-to-donor variability, enhanced MSC immunoregulatory capacity, and better differentiation potential [18]. However, the selection of a specific SF medium for MSC therapy still requires careful consideration to align with targeted therapeutic objectives [19]. Often, the choice of an SF culture medium is driven mainly by the medium’s ability to support high cell productivity [18]. While there may be a correlation between the growth potential and the general cell quality of MSCs [20], the selection criteria should be re-evaluated for specific clinical applications of MSCs in targeted tissue. Previous studies from our team revealed that multipotent MSCs (with the highest growth potential), identified by unique biophysical attributes [21], were not necessarily the best cells for in vivo efficacy [22,23]. For application in cartilage regeneration, the chondrogenic potential of MSCs should be prioritized over their proliferative capacity in assessments [24].

Unfortunately, there is a notable gap in the literature in terms of rigorous evaluation of MSC chondrogenic potential following subjection to SF expansion. As listed in Table 1, 10 out of 40 serum/xeno-free MSC characterization studies did not include chondrogenic potential validation. For most MSC chondrogenesis assessments, the outcome is not conclusive due to a lack of discriminative controls or defined endpoints. In addition, 15/27 of the 3D chondrogenic or high-density micromass induction assessments lacked histomorphometry quantification, and 2/27 of these studies did not perform any extracellular matrix (ECM) staining. Moreover, 4/27 studies performed a 2D induction protocol, which has been recognized to not consistently reflect MSC chondrogenic potential [25].

To demonstrate the empirical need to qualify the ability of an SF medium to support the chondrogenic potency of expanded MSCs for cartilage repair, we conducted a comprehensive assessment of MSCs expanded in two SF media in comparison to FBS-media-expanded MSCs. The two tested SF media included a commercially marketed MesenCult™-ACF Plus medium (ACF), which is a continuation of the MesenCult™-XF medium from Stem Cell Technologies, and a custom MSC serum/xeno-free medium (XF) newly developed by ThermoFisher. Assessments include cell morphology, proliferation rate, cell surface markers, trilineage differentiation potential, and cellular activities. To qualify as having a chondrogenic ability, expanded MSCs were subjected to 3D aggregate induction followed by ECM staining and quantification. The in vivo cartilage repair efficacy of MSCs was investigated using a rat osteochondral defect model.

## 2. Results

### 2.1. In Vivo Experiment

#### 2.1.1. Histological Assessment of Cartilage Repair

The cartilage repair efficacy of MSCs expanded in MesenCult™-ACF Plus SF medium (ACF-MSC) was tested using a rat osteochondral defect model in reference to 10% FBS DMEM expanded MSC (FBS-MSC). ACF-MSCs and FBS-MSCs derived from two donors were implanted and encapsulated in hydrogel. Cell-free hydrogel (Hydrogel) served as the control.

FBS-MSCs from both donors showed obvious hyaline cartilage regeneration, with the defect area repaired with tissues composed mainly of COLII and aggrecan deposition (Figure 1B). The FBS-MSC groups augmented tissue repair with an overall modified O’Driscoll score that was significantly higher than the Hydrogel group (Figure 1C,F). In contrast, implantation with ACF-MSCs from both donors did not yield improvement relative to the Hydrogel group.

#### 2.1.2. Micro-CT Assessment of Subchondral Bone Repair and Mechanical Strength of Repair Tissue

Among the four groups, FBS-Donor 2 showed the best subchondral bone regeneration, with more than three-quarters of the defect area filled with regenerated tissues (Figure 2A). In addition, micro-CT analysis revealed significant improvement in the FBS-Donor 2 group in terms of regenerated bone volume (Figure 2B), higher trabecular number (Figure 2C), and decreased trabecular separation distance (Figure 2D) compared with the hydrogel control and the ACF-Donor 2.

Although no significant structural improvement was found for the other three groups, the compression strength test revealed functional enhancement of regenerated tissues for all four MSC-transplanted groups compared with the hydrogel control. Furthermore, 2/6 of ACF-Donor 1, 2/6 of FBS-Donor 1, 1/6 of ACF-Donor 2, and 3/6 FBS-Donor 2 showed strength comparable to that of the healthy control.

### 2.2. In Vitro Experiment

#### 2.2.1. SF Medium Reduced Donor–Donor Variability in Terms of Growth Rate and Cell Morphology

MSCs from the donors were expanded in 10% FBS, ACF, or XF medium from P3 to P5. Three typical MSC morphologies were observed: a long-fibroblast-like spindle-shaped MSC (SS), a small star-shaped MSC (RS), and a large, flattened cuboid-shaped MSC (FC) [64] (Figure 3A).

At P3, distinctive cell morphologies were observed for each group (Figure 3A). In FBS culture, cells were the most heterogeneous, consisting of SS, RS, and FC morphologies with larger cell sizes (mean cell size: 17.07–19.59 µm) (Figure 4C). In the ACF culture, the majority of the cells from all three donors resemble an RS morphology (mean cell size: 15.05–15.77 µm) (Figure 3A and Figure 4C). In the XF cultures, the cells were less heterogeneous and smaller than in the FBS cultures (mean cell size: 16.52–17.26 µm) (Figure 4C) but consisted of two distinctive cell morphological subpopulations of SS and RS cells (Figure 3A). The suspension cell size distribution showed that the SF expansion culture maintained a smaller cell size and cell population homogeneity for all three donors (Figure 4C).

At P5, there was an elevation in both MSC heterogeneity and suspension cell size for MSCs in all three medium cultures, with the exception of FBS-Donor 2 (Figure 3B and Figure 4C). In the FBS culture, Donors 1 and 3 showed a higher proportion of FC cells.

In the FBS culture, Donor 2 exhibited higher growth rates than the other two donors at P3. However, both SF expansions resulted in faster growth relative to the FBS condition, with equivalent growth rates across donors. ACF-MSCs have the highest growth rates, exceeding those of FBS-MSCs by more than threefold at P3, while XF-MSCs nearly doubled their growth rates (Figure 4A). By P5, growth had reduced in all groups except for FBS-Donor 2 (Figure 4B). With the exception of Donor 2, MSCs in both SF conditions expanded faster, with ACF again supporting the fastest growth.

#### 2.2.2. SF Medium Preserved Expression of MSC Characterization Markers

The expression of MSC characterization surface markers was analyzed by FACS (Table 2). Although ACF-MSCs preserved a high percentage of MSC-positive markers, there was a significant decrease in fluorescence intensity of CD105 and CD90 for all three donor cells compared with the FBS culture. A similar decrease in CD105 expression was found for XF-MSCs. In addition, a significant reduction in CD146 positive cell percentage and fluorescence intensity was also found in XF-MSCs, suggestive of a possible MSC lineage deviation.

#### 2.2.3. MSC Morphology at Higher Confluency and Later Passage

An obvious clustered feature began to form in ACF-MSCs at P5, where the cells exhibited a reduced degree of contact inhibition and started to grow in layers, one on top of the other. This phenomenon was not observed in other groups of cells, even at a high density of 90% confluency (Figure 5A). Large vacuoles were also observed in the ACF-MSCs’ cytoplasm (Figure 5A).

At Passage 6 (P6), cell proliferation halted for ACF-MSCs. A more pronounced clustering of cells was observed on day 8 of cell culture, along with the development of an elongated, hypertrophic-like cell morphology (Figure 5B). This phenomenon was not observed in other media.

These morphological changes were consistent across all three donors; Figure 5 shows their representative morphology with Donor 2.

#### 2.2.4. Trilineage Differentiation Assay

After two passages of expansion culture, MSCs were harvested by the end of P5 for trilineage differentiation.

For chondrogenesis, within the FBS group, Donor 2 showed moderate chondrogenic differentiation with partial COLII staining in the tissue pellet (Figure 6A). ACF-MSCs performed poorly in chondrogenesis differentiation, with the absence of COLII expression (Figure 6A) in all three donors. On the contrary, the XF medium enhanced the MSC chondrogenesis potential in all three donors with full COLII expression (Figure 6A). These trends were also reflected in the formation of sulfated glycosaminoglycans quantification (sGAG) (Figure 6B).

For osteogenesis, the XF and ACF groups showed elevated levels of calcium deposition (Figure 6A) and increased alizarin stain quantification (Figure 6B) compared with the FBS group.

For adipogenesis, the ACF group showed elevated oil droplet formation compared with the FBS group (Figure 6A,D). The adipogenesis outcome in the XF group was equivalent to that of the FBS group (Figure 6A,D).

#### 2.2.5. Comparison of MSC Proliferation, Senescence, and Apoptosis Markers

SA-beta-Gal staining revealed a significantly higher senescence level for the FBS group than the two SF groups. Within the FBS group, Donor 2 was carrying a significantly lower extent of senescence cells (Figure 7A), senescence marker P16, and gene expression (Figure 7D). ACF-MSCs had the lowest levels of senescence in both beta-gal staining and P16 expression (Figure 7A,D). The presence of low levels of senescence cells relative to FBS-MSCs was detected in XF-MSCs. The elevation in the percentage of beta-gal-positive cells in XF-Donor 3 was in accordance with the increase in FC cells observed in the cell culture (Figure 3B and Figure 7A). Nevertheless, all three donors showed high levels of P16 at P5, in accordance with their enlarged cell size (Figure 3B and Figure 7D) [22].

The FBS group exhibited a lower expression of proliferation marker Ki-67 compared with the SF groups (Figure 7B). FBS-Donor 2 maintained a moderate Ki-67 index of 18.3%, correlated to its maintenance of proliferative capacity at P5 (Figure 4B). In general, the Ki-67 expression levels across the MSC donors under different expansion conditions correlated to their proliferation rate.

For apoptosis, the ACF group showed increased caspase 3/7 expression. Although no significant change in P53 and P21 was detected at P5, compared with the FBS group, an elevated P53/P21 ratio was observed in Donors 2 and 3 (Figure 7E). The caspase 3/7 expression in XF-MSCs was similar to FBS-MSCs. Although P53 and P21 were elevated at P5, the P53/P21 ratio was lower (Figure 7E–G).

## 3. Discussion

While all commercialized SF media state more stable clinical-grade cell production, each SF medium is unique in its formulation and, thus, has its own advantages and disadvantages. As a result, medium testing is essential to ensure quality cell production for specific clinical applications. Although SF media outperform in terms of cellular proliferation and senescence inhibition, this does not necessarily translate to better differentiation potential or in vivo performance. Of the two tested SF MSC media, the ACF medium supported high MSC proliferation, resulting in expanded MSCs with enhanced adipogenic and osteogenic potential; however, they lost their chondrogenic ability. The XF medium, on the other hand, had a more moderate proliferative effect while maintaining the trilineage potential of MSCs and, in fact, heightened the chondrogenic and osteogenic potential of the expanded MSCs.

The application of an SF medium often requires complementary SF cell attachment substrates, which affect the adhesion properties of MSCs and could alter the expression of MSC surface glycoproteins and their subsequent downstream signaling, especially in long-term cultures [44]. Characterization of surface markers, CD90, CD105, and CD146, which are involved in cell angiogenesis [65,66] and cell–cell adhesion pathways [67], of the expanded MSCs revealed subtle differences. Consistent with our results, reductions in CD90 and CD105 expression have been reported for hBMSCs cultured in Mesencult-XF medium [44,68], and reduction in CD90 expression was reported by Cimino et al. (2018) for StemPro™ MSC SFM [69].

CD105, also known as TGF-beta receptor III, participates in the TGFβ/Smad 2 pathway in promoting chondrogenesis [70]. Down-regulation of the TGFβ/Smad 2 pathway has been shown to increase hBMSC colony formation, migration, and invasion, accompanied by elevated cell adipogenic [71] and osteogenic potential [72]. Similarly, decreased CD90 expression is associated with increased adipogenesis and osteogenesis potential [73]. The increased adipogenic and osteogenic potential concomitated with the loss of chondrogenic ability of ACF-MSCs could thus be associated with reduced CD105 and CD90 expression. While XF-MSCs also experienced reduced CD105 expression, lost or reduced CD146 expression was detected. Down-regulation of CD146 leads to decreased activation of Wnt/β-catenin signaling, allowing MSCs to enter chondrogenesis more readily [74,75], thus accounting for the increased chondrogenic potential of XF-expanded MSCs.

Despite alleviating senescence progression, as indicated by lack of β-gal staining and elevated P16 expression (Figure 7), our results showed that ACF medium inhibits MSC chondrogenic differentiation (Figure 6A,B). P16 often indicates a permanent state of cellular senescence due to its role in senescence maintenance [76], while P53 and P21 have been reported to initiate the MSC senescence pathway. Nevertheless, P53, P21, and P16 form a complicated trajectory for cellular proliferation, senescence, apoptosis, transformation, and DNA repair [77]. P21, as a downstream effector of P53, often plays the role of triggering DNA damage repair (DDR), during which P21 initiates a cellular arrest state to facilitate DDR or senescence initiation. On the other hand, extensive activation of P53 often leads to direct cell apoptosis, which is correlated with a high P53:P21 ratio [78]. A previous study reported that early chondrogenesis events trigger pro-apoptotic pathways, which impair MSC survival and in vivo engraftment outcomes [79]. In this study, the P53:P21 ratio was elevated in ACF-MSCs while apoptosis was increased, as indicated by higher caspase 3/7 activity (Figure 7). XF-MSCs, on the other hand, have repressed senescence without the elevation of apoptosis. The low P53:P21 ratio suggests pro-survival DDR repair pathway activation.

ACF-MSCs predominantly consist of RS cells that are proposed to be self-replicative but harness low chondrogenic potency [80], while XF-MSCs consist of both RS and SS cells, which are correlated with better differentiation functions [80]. Notably, we observed the formation of large vacuoles within the cytoplasm of ACF-MSCs at P5 and P6, which was reported by Al-Saqi et al. (2014) to be senescent and found in late-passage hBMSC cultured in MesenCult-XF [44]. Similar early observations of fatty vacuole formations have been reported in FBS-MSCs experiencing senescence and epithelial transformation [81]. The accumulation of lipofuscin, an autofluorescent lipophilic structure, was established as a phenotype for early senescence [82]. However, our experiment showed that ACF-MSCs did not stain positive for either lipophilic Oil red O or senescent beta-gal assay, indicating that the vacuolar structures formed (Figure 5A) are not of lipophilic origin and require further investigation. In our studies, although ACF repressed the senescence phenotype and enhanced proliferation, a halt in growth was observed in ACF, as well as the XF medium, for later-passage MSCs at P6 and P7, respectively. Moreover, the capacity of the SF medium to maintain MSC growth rates and reduce population heterogeneity decreased progressively across each passage. It has been suggested that although the SF medium can increase the growth rates of MSCs, there still remains a limited capacity for population doubling [83].

To replace the serum in culture medium, SF mediums are often enriched with a specific formulation of growth factors such as fibroblast growth factors (FGFs), epidermal growth factors (EGFs), transforming growth factors (TGFs), vascular endothelial growth factors (VEGFs) and platelet-derived growth factors (PDGF) [84]. Exogenous supplementation of these growth factors has been proven to increase MSC proliferation. However, their long-term utilization has been shown to promote senescence in later passages. Contradictory reports on the TGFβ family revealed their dose- and phase-dependent regulation on MSC senescence [85]. As demonstrated by Mazzella, Walker, Cormier, Kapanowski, Ishmakej, Saifee, Govind, and Chaudhry [1], TGFβ signaling is responsible for the P53-independent activation of P21 and senescence induction in UCMSCs. In contrast, FGF2 and PDGF have been shown to decrease the proliferative property of MSCs in late passage [86] via the induction of the SASP secretome. The rise in senescence markers in XF-MSCs might be due to the senescence induction effect from growth factor enrichment, potentially via the TGFβ/Smad2 pathway, which primed the cells for chondrogenic induction.

Taken together, we found that despite the promotion of MSC proliferation by the SF medium, the expanded MSCs could constitute variations in morphology, surface markers, senescence status, and differentiation capacity. Each SF medium can vary widely, and the precise formulation is proprietary information, with minimal evidence regarding their specific functional potential, which could be responsible for the unsatisfactory outcome in a substantial number of clinical trials. When considering employing an SF medium in MSC expansion for clinical application in cartilage repair, the users need to perform a thorough investigation on the chondrogenic potency of expanded MSCs, such as the in vitro 3D chondrogenic induction method, coupled with ECM staining and quantification. This study contributes to addressing the current gap in SF media applications, while more preclinical validations will still be in demand in the future.

## 4. Materials and Methods

### 4.1. MSC Cell Culture

Bone-marrow-derived MSCs (BMSCs) from 3 different donors, as described in Table 3, were purchased from Lonza Pte. Ltd. and Stem Cell Technologies (Vancouver, BC, Canada). The cells were expanded in tissue culture plate (TCP) at density of 1500 cells/cm^2^ in low glucose Dulbecco’s Modified Eagle Medium (DMEM) supplemented with 10% fetal bovine serum (FBS), 1% GlutaMAX, and 1% Penicillin/Streptomycin (Thermo Fisher Scientific, Singapore, Singapore) at 37 °C in 5% CO_2_ atmosphere. The medium was changed every 2 days, and the cells were harvested at 80% confluency for further experiments or subcultures. For serum culture control, 10% FBS-MSC culture medium was employed as described above, with medium changed every 3–4 days. For SF ACF culture, cell culture flasks were first coated with Animal Component-Free Cell Attachment Substrate (Stem Cell Technologies, Vancouver, Canada) at room temperature for 2 h. Cells were cultured in MesenCult™-ACF Plus Medium (Stem Cell Technologies, Vancouver, BC, Canada), with medium changed weekly. For SF XF culture, cells were directly seeded to Nunclon™ Supra Surface flasks (156374, ThermoFisher, Singapore, Singapore) without attachment substrate. Cells were cultured in Custom MSC SF XF Medium Kit (ME20236L1, Gibco, New York, NY, USA), consisting of StemPro™ MSC SFM Basal Medium and Custom MSC SF XF Supplement. Additional growth factors were supplemented as recommended by manufacturer’s protocol to final concentration of PDGF-BB (20 ng/mL), FGF basic (4 ng/mL), and TGFβ1 (0.5 ng/mL). All seeding densities were controlled at 2000 cells/cm^2^. During cell harvest, MSCs were incubated in TrypLE (Thermo Fisher Scientific, Singapore) for 5 min. MSCs were cultured in these 3 conditions for 2 passages and were harvested at Passage 4 for animal experiment and subsequent biomarker and functional analysis.

### 4.2. Animal Experiment

All procedures were performed according to the guidelines of the Institutional Animal Care and Use Committee at National University of Singapore (protocol No. R20-0258). Female Sprague Dawley rats (12 weeks old, weight range 300–400 g) were used. Osteochondral defect (1.5 mm diameter, 1 mm depth) was created manually on the trochlear grooves of rat distal femurs with a drill. The defects in both knees were implanted with a cell-free fibrin hydrogel (control group; *n* = 8), MSCs expanded in MesenCult™-ACF SF medium (SF-MSC group; *n* = 6), MSCs expanded in 10% FBS DMEM (FBS-MSC group; *n* = 6). MSCs were mixed with a fibrin hydrogel at 10 million cells/mL, with 60,000 cells implanted into each defect. Rats were given cyclosporine-supplemented water (35 mg/kg, Novartis, Basel, Switzerland) to prevent host rejection of the implanted human cells. Eight weeks after surgery, the rats were euthanized by CO_2_ inhalation. Distal femora were harvested for compression assessment or fixed in 10% buffered formalin. Formalin-fixed tissue was subjected to micro-computed tomography (micro-CT) before undergoing histological processing, sectioning, and immunohistochemical analyses. The quality of cartilage repair was assessed with the modified O’Driscoll histological scoring system (Appendix A) by 3 blinded independent researchers.

### 4.3. Mechanical Strength Test

Eight weeks after surgery, the operated knees were subjected to biomechanical testing using the Instron 5543 machine (Instron, Singapore, Singapore) equipped with a 10 N load cell. Briefly, the distal end of the femur was fixed using a custom-made rigid clam. A circular indenter with a 1 mm diameter tip was positioned perpendicular to the defect area for indentation and compression testing. The loading rate was set at 0.01 mm/min until 15% compressive strain was reached. Young’s modulus in megapascals was derived from the first linear part of the stress versus strain data.

### 4.4. Micro-Computed Tomography (Micro-CT)

The distal femurs were fixed in 10% buffered formalin for a week before micro-CT using a micro-CT scanner (Bruker, Billerica, MA, USA). Each defect was scanned with a resolution of 40 mm/pixel. A rectangular region of interest perpendicular to the defect surface (1.5 mm in width and 1.5 mm in height), positioned in the middle of the defect, was selected on 20 continuous slides and regarded as the total volume. The bone volume fraction (BV/TV; measured in percent), trabecular thickness (Tb.Th; measured in millimeters), trabecular separation (Tb.Sp; measured in millimeters), and trabecular number (Tb.N; 1/mm) were calculated by CT-Analyser (Skyscan, Bruker, Billerica, MA, USA).

### 4.5. Histological and Immunohistochemical Staining

The distal femurs were fixed in 10% neutral buffered formalin (Sigma-Aldrich, Singapore, Singapore) for 1 week, followed by decalcification in 30% formic acid for 2 weeks, with weekly change of decalcification solution. MSC-differentiated chondrocyte pellets were fixed for 24 h. Samples then underwent ethanol dehydration, paraffin embedding, and sectioning into 5 µm slides. The general structure was stained by Eosin (Sigma-Aldrich). Proteoglycan was stained with 0.1% Safranin O solution (Acros Organics, Geel, Belgium) and counterstained with 0.02% Fast green solution (Sigma-Aldrich). Immunohistochemical staining was performed with Type II collagen (Col2) mouse monoclonal antibodies (Clone 6B3, 1:500 dilution, Chemicon, Rolling Meadows, IL, USA). Endogenous peroxidase was first blocked using hydrogen peroxide, and the sections were digested with pepsin to retrieve antigens. After blocking, the sections were then incubated in primary antibody and secondary antibody (replacement) for 1 h and 10 min, respectively, at room temperature. Staining visualization was revealed by DAB chromogen and substrate mix application after 15 min incubation in Strep peroxidase solution (replacement). All slides were counterstained with Accustain^®^ Harris hematoxylin (Sigma-Aldrich, HHS32).

### 4.6. Trilineage Differentiation

For adipogenesis and osteogenesis induction, cells were seeded into 24 well plates at densities of 4 × 10^5^ cells per well and 6 × 10^5^ cells per well respectively. The cells were allowed for attachment in 10% FBS low glucose DMEM for 24 h. The medium was then replaced with respective differentiation media. The adipogenesis medium contained 10% FBS, 10 µM dexamethasone, 0.01 mg/mL insulin, 0.5 mM isobutylmethylxanthine, 200 µM indomethacin, 1% penicillin–streptomycin and 2 mM GlutaMAX in 4.5 g/L D-glucose DMEM (Thermo Fisher Scientific). The osteogenesis medium contained 10% FBS, 10^−7^ M dexamethasone, 50 µg/mL ascorbic acid, 10 mM β-glycerophosphate, 1 mM sodium pyruvate, 1% penicillin–streptomycin, and 1 mM GlutaMAX in 1 g/L D-glucose DMEM (Thermo Fisher Scientific). The cells were cultured for 2 weeks with change of media on every alternate day. Extent of adipogenesis and osteogenesis was quantified by Oil red O staining (Sigma-Aldrich) and Alizarin staining (Sigma-Aldrich), respectively.

For chondrogenesis induction, cell pellets were first formed by centrifugation of 0.15 × 10^6^ cells per well in chondrogenesis medium at 300 g for 5 min in low-attachment 96-well plate. The culture medium, consisting of 10^−7^ M dexamethasone, 50 mg/mL ascorbic acid, 4 mM proline, 1% ITS Premix supplement (Corning), 1 mM sodium pyruvate, 1% penicillin–streptomycin, 1 mM GlutaMAX, and 10 ng/mL transforming growth factor–β3 in 4.5 g/L D-glucose DMEM (Thermo Fisher Scientific), was changed every alternate day for 3 weeks. Differentiated cell pellets were harvested for extracellular matrix quantification (qECM) and histological staining.

### 4.7. Real-Time Polymerase Chain Reaction (qPCR)

Total RNA was extracted with RNeasy^®^ Mini Kit (Qiagen, 74104, Singapore, Singapore), and reverse transcription was performed with iScript^TM^ cDNA synthesis kit (Bio-Rad). Real-time PCR was carried out with POWER SYBRR^®^ green PCR master mix on ABI 7500 Real-time PCR System (Applied Biosystem, Foster City, CA, USA). The gene expression level was normalized to glyceraldehyde-3-phosphate dehydrogenase (GAPDH) and was calculated using the formula 2^−∆∆Ct^. Primer sequence are listed in Table 4.

### 4.8. Flow Cytometry

Passage 5 MSCs were used for cell surface marker (FACS) analysis. Cells were individually incubated with fluorophore-conjugated antibodies of CD44, CD105, CD90, and CD146. Flow cytometry was performed on CytoFLEX S Flow Cytometer (Beckman Beckman Coulter, Brea, CA, USA).

### 4.9. Ki-67 Immunofluorescence Staining

MSCs were seeded in 48-well plates at density of 10,000 cells per well and allowed to attach overnight. Cells were then conjugated with Ki-67 primary antibody (MA5-14520, Thermo Fisher Scientific), followed by Alexa 488 conjugated donkey anti-mouse secondary antibody (A21202, Invitrogen^TM^, Singapore, Singapore). Microscopy scans were taken with OLYMPUS IX81.

### 4.10. Beta-Gal Staining

Senescence Cells Histochemical Staining Kit (+CS0030, Sigma Aldrich, Darmstadt, Germany) was used to quantify the extent of senescence according to the manufacturer’s protocol. Briefly, cells were seeded to 48-well plate at density of 10,000 cells per well. After 24 h, the cells were fixed and incubated in staining mix containing β-gal. Five random areas were captured at the bright field by a color camera for quantifying the β-galactosidase staining-based senescence index (β-gal Sn index) normalized by total cell confluency.

### 4.11. Caspase-3/7 Assay

Caspase-3/7 activity was quantified by the Caspase-Glo^®^ 3/7 assay (G8090, Promega, Fitchburg, WI, USA) according to manufacturer protocol. Briefly, cells were seeded in 96-well plates at density of 5000 cells per well. After 24 h, cells were equilibrated to room temperature and incubated with Caspase-Glo^®^ 3/7 Reagent for 1 h. Luminescence readings were then taken by Infinite^®^ 200 PRO (Tecan, Mannedorf, Switzerland.).

### 4.12. Image and Statistical Analysis

All image analysis was performed on imageJ (version 1.54). Statistical analysis was carried out with 2-tail Welch’s *t*-test for comparison between 2 groups on Prism GraphPad, with significance set at *p* < 0.05. Data are presented as mean ± SD. Histological analysis was performed by 3 investigators blinded to sample identity.

### 4.13. Selection of Articles for Review Summary

Articles were filtered from PubMed database with the following query (((“MSC”) OR (“Mesenchymal stem cell”) OR (“Mesenchymal stromal cell”)) AND ((“serum free”) OR (“serum-free”) OR (“xeno-free”))) NOT ((Pluripotent stem cell) OR (Exosome) OR (secretory) OR (extracellular vesicles) OR (secrete) OR (extracellular matrix) OR (virus) OR (COVID-19)). Year filter was added from 2011 to 2024. Out of the 133 articles returned, 93 articles were excluded for non-related or non-open-access content and reviews. A total of 40 articles were selected for the review summary.

## Figures and Tables

**Figure 1 ijms-25-10627-f001:**
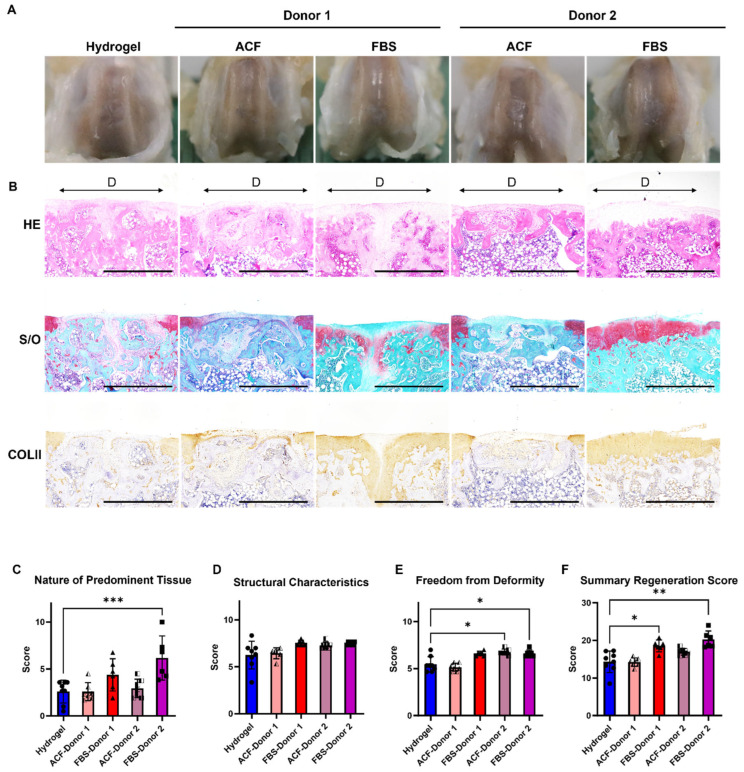
Evaluation of in vivo osteochondral repairs. (**A**) Macroscopic appearance of osteochondral defects at 6 weeks post-surgery. (**B**) Histological and immunohistochemical staining of the repaired cartilage. Hyaline cartilage was stained red in Safranin O staining and brown color in Collagen II immunobiological staining. Scale bar = 1 mm. HE = Hematoxylin and Eosin; S/O = Safranin O; COLII = Collagen II; D = Defect. (**C**–**E**) The modified O’Driscoll score for cartilage regeneration with three parameters, including nature of predominant tissue, structural characteristics, and freedom from deformity. (**F**) Overall histology score. * = *p* < 0.05; ** = *p* < 0.01; *** = *p* < 0.001.

**Figure 2 ijms-25-10627-f002:**
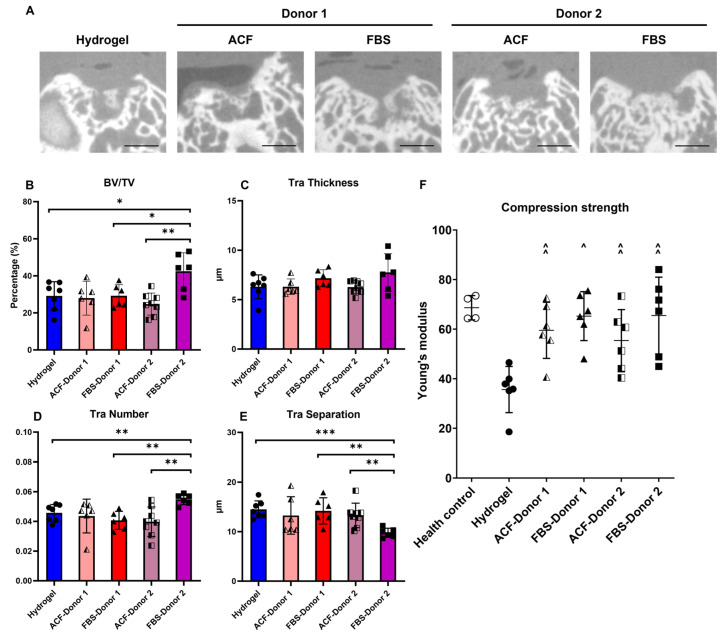
In vivo study: structural and mechanical functionality of ACF and FBS-MSCs repaired cartilage. (**A**) Middle cross-section of micro-CT scan for repaired defects. Scale bar = 1 mm. (**B**–**E**) Micro-CT quantitative analysis of regenerated subchondral bone. BV = Bone Volume; TV = Tissue Volume; Tra = Trabecular. * = *p* < 0.05; ** = *p* < 0.01; *** = *p* < 0.001 between-pair groups. (**F**) Compression strength of repaired defect area. ^ = *p* < 0.05, ^^ = *p* < 0.01 compared with Hydrogel group.

**Figure 3 ijms-25-10627-f003:**
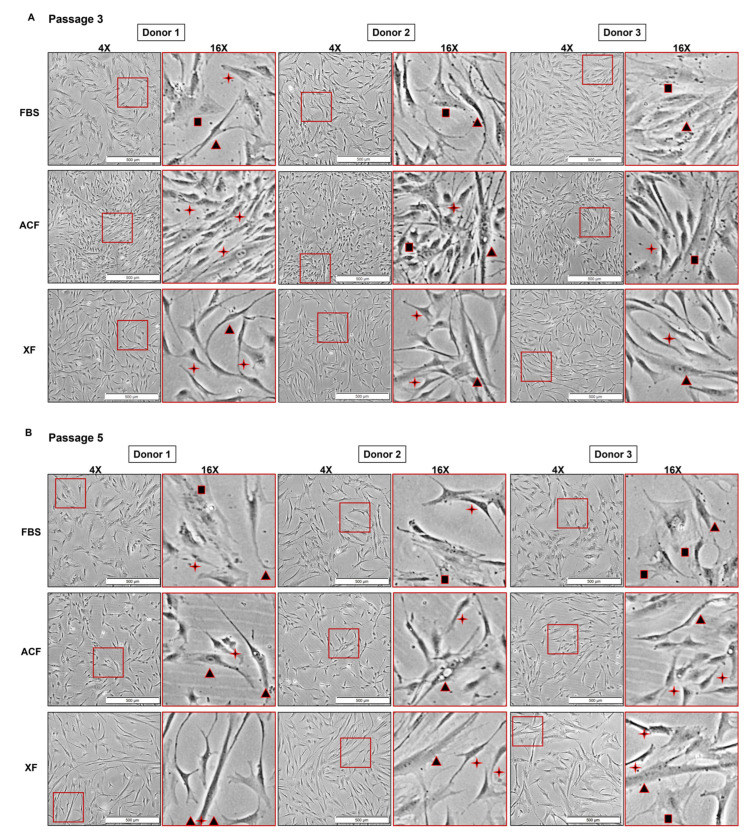
Morphology of MSCs at Passage 3 (P3) and Passage 5 (P5). (**A**,**B**) Cell morphology at P3 and P5, respectively; cell confluency at 50–60%, scale bar = 500 µm. ✦: small star-shaped MSC (RS); ▲: long-fibroblast-like spindle-shaped MSC (SS); ■: flattened cuboid-shaped MSC (FC).

**Figure 4 ijms-25-10627-f004:**
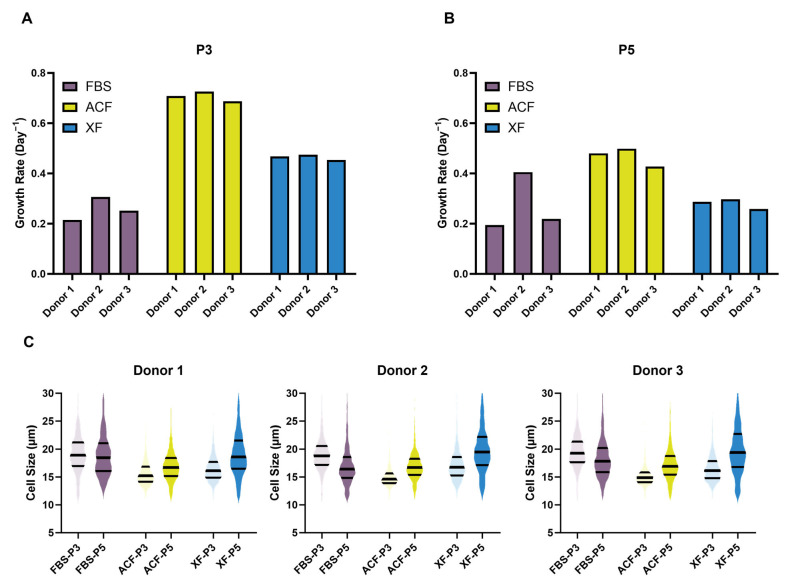
MSC growth rates and suspension cell size. (**A**,**B**) Growth rate at P3 and P5, respectively. (**C**) MSC suspension cell size, median, and quartile cell size are indicated by the black line.

**Figure 5 ijms-25-10627-f005:**
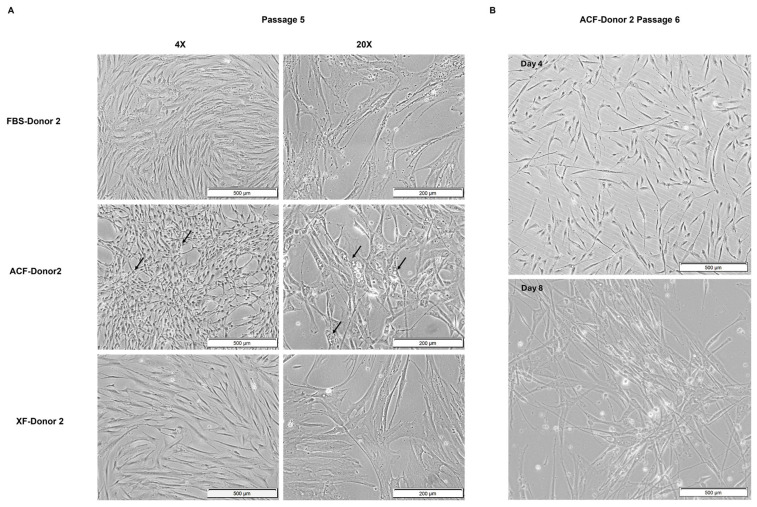
Abnormality in cellular morphology for ACF-MSCs, represented by Donor 2 cells. (**A**) P5 MSC morphology at 60–90% confluency. ACF-MSCs typically showed large vesicle formation. (**B**) P6 ACF-MSCs on day 4 and day 8, magnification = 4×. Black arrow indicates vacuolar structures.

**Figure 6 ijms-25-10627-f006:**
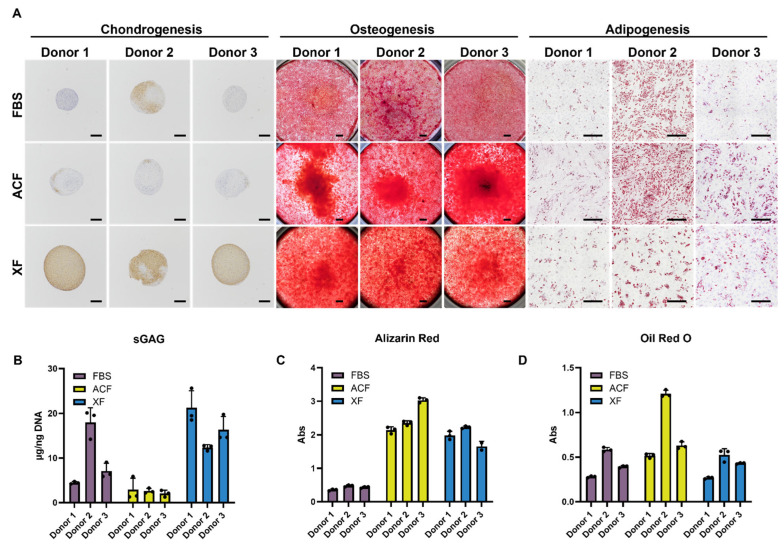
Comparison of serum and SF media on trilineage differentiation functions. (**A**) COL II staining of chondrogenic pellet, magnification = 4×, scale bar = 200 µm; osteogenesis and adipogenesis assay images were taken by scanning 3 × 3 field large image with magnification of 4× (scale bar = 1 mm) and 20× (scale bar = 500 µm), respectively. (**B**) sGAG quantification from digested chondrogenic pellets. (**C**,**D**) Alizarin red and Oil red O quantification via color extraction from stained wells, respectively.

**Figure 7 ijms-25-10627-f007:**
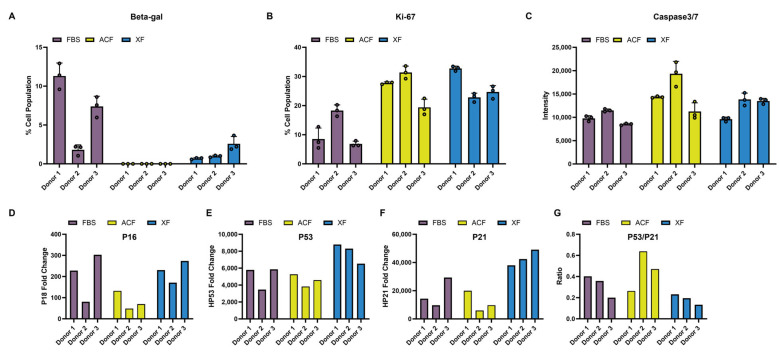
Comparison of senescence and cell cycle regulation markers in Passage 5 MSC. (**A**) Quantitative analysis of SA-beta-Gal staining of senescent MSCs, represented by % cell population. (**B**) Ki-67 proliferation index. (**C**) Caspase 3/7 quantification. (**D**–**F**) Gene expression normalized to GAPDH expression for P16, P53, and P21, respectively. (**G**) P53:P21 ratio.

**Table 1 ijms-25-10627-t001:** Summary review on MSC SF medium studies dated since 2011. NA: No relevant analytical assay performed or not mentioned; -: equivalent result for SF and FBS control, non-conclusive comparison or no comparable control group performed; ↑: increased potential; ↓: decreased potential. Abbreviations: Osteo—Osteogenesis; Adipo—Adipogenesis; MSC—Mesenchymal stromal cell; hBFP—Human buccal fat pad; hLP—Human periodontal ligament; hDP—Human dental pulp; hBMSC—Human bone-marrow-derived MSC; hADSC—Human adipose-derived stem cell; hUCSC—Human umbilical-cord-derived stem cell; hFCPC—Human fetal cartilage-derived progenitor cell; mPMSC—Mouse palatal-derived MSC; hPL—Human platelet lysate, hCMSC—Human cartilage-derived MSC; hPLMSC—Human palate-derived MSC; hPDSC—Human placenta-derived stem cell; HESC—Human endometrial stem cell, hWJMSC—Human Wharton’s Jelly-derived MSC.

SF/XF Medium	MSC Source	MSC Characterization Marker	Chondrogenesis			Osteo	Adipo	Ref.
			Trend	Method	End-Point			
StemPro™ MSC SFM, Gibco	hADSC	+ve: CD73, CD90, CD105 −ve: CD11b, CD19, CD34, CD45, HLA-DR	-	3D aggregate chondrogenic induction, followed by histological alcian blue staining	21 days		-	[26]
	hBFP, hLP, and hDP derived MSCs	+ve: CD73, CD90, CD105 −ve: CD11b, CD19, CD34, CD45, HLA-DR	NA	NA	NA	↑	NA	[27]
	hBMSC, hADSC, hUCSC	+ve: CD73, CD90 and CD105 −ve: CD14, CD19, CD31, CD34, CD45, CD80, HLA-DR	-	3D aggregate chondrogenic induction, brightfield image examination	14 days	-	-	[28]
	hUCSC	+ve: CD73, CD90, CD105 −ve: CD31, CD80, HLA-DR	-	3D aggregate chondrogenic induction, followed by alcian blue staining	15 days	-	-	[29]
	hADSC, hBMSC	+ve: CD73, CD90, CD105 −ve: CD31, CD80, HLA-DR	-	High-density micromass cultures, followed by alcian blue staining	14 days	-	-	[30]
	hFCPC	+ve: CD44, CD73, CD90, CD105, CD166 −ve: SSEA-4 and TRA-1-60	-	3D aggregate chondrogenic induction, followed by histological safranin O staining	14 days	-	-	[31]
	hBMSC	+ve: CD73, CD90, CD105, CD146 −ve: CD14, CD19, CD34, CD45, HLA-DR	↑	High-density micromass cultures, followed by alcian blue histological staining	21 days	-	-	[32]
	hBMSC	+ve: CD73, CD90, CD105 −ve: CD14, CD19, CD34, CD45, HLA-DR	↑	3D aggregate chondrogenic induction, followed by immunohistochemical COLII staining	14–20 days	-	-	[33]
CellCor, X Cell Therapeutics; StemPro™ MSC SFM, Gibco; Mesencult™-XF, Stem Cell Technologies	hBMSC, hADSC, hUCSC	+ve: CD73, CD90, CD105, CD146 −ve: CD14, CD34, CD45, HLA-DR	-	2D induction culture followed by alcian blue staining	NA	↑	-	[18]
E8 medium, ThermoFisher	hPLMSC	+ve: CD73, CD90, CD105	-	High-density micromass cultures, followed by alcian blue staining	16 days	-	-	[34]
MSC NutriStem^®^ XF Medium, Biological Industry	mPMSC	+ve: CD29, CD44, CD90 −ve: CD34, CD45, CD146	-	3D aggregate chondrogenic induction, followed by alcian blue staining	7 days	-	-	[35]
MSC NutriStem^®^ XF Medium + 0.5% hPL, Biological Industry	hCMSC	+ve: CD73, CD90, CD105	↑	3D aggregate chondrogenic induction, followed by alcian blue staining	14 days	↑	↑	[25]
RoosterNourish XF/Low Serum, RoosterBio; StemMACS™ MSC Expansion Media XF, Miltenyi; PLTMax Human Platelet Lysate, Sigma; StemXVivo™, Lonza	hBMSC	+ve: CD73, CD90, CD105 −ve: CD34, CD45, HLA-DR	-	3D aggregate chondrogenic induction	NA	-	-	[36]
PRIME-XV MSC Expansion XSFM medium, FUJIFILM Irvine Scientific	hBMSC	+ve: CD44, CD73, CD90, CD105, CD146, CD166 −ve: CD11b, CD14, CD34, CD45,	-	2D induction culture followed by alcian blue staining	28 days	-	-	[37]
	hBMSC	+ve: CD73, CD90, CD105 −ve: CD34, HLA-DR	-	3D aggregate chondrogenic induction, followed by alcian blue staining	21 days	-	-	[38]
	hBMSC	+ve: CD73, CD90, CD105 −ve: CD34, HLA-DR	-	3D aggregate chondrogenic induction, followed by alcian blue staining	21 days	-	-	[39]
PowerStem MSC1 Medium, PAN Biotech; StemMACS™ MSC Expansion Media XF, Miltenyi	hBMSC	+ve: CD73, CD90, CD105 −ve: CD34, HLA-DR	-	3D aggregate chondrogenic induction, followed by alcian blue staining	21 days	-	-	[39]
MSC-Brew GMP Medium, Miltenyi	hBMSC	+ve: CD73, CD90, CD105, and CD146 −ve: CD34, CD45, HLA-DR	NA	NA	NA	↑	↑	[40]
StemMACS™ MSC Expansion Media XF, Miltenyi; PLTMax Human Platelet Lysate, Sigma; MesenCult-hPL media, StemCell Technologies	hADSC	+ve: CD73, CD90, CD105	NA	NA	NA	-	-	[41]
BD Mosaic™ Mesenchymal Stem Cell SF media, ThermoFisher; MesenCult™-XF, Stem Cell Technologies	hBMSC	+ve: CD73, CD90, CD105, CD166 −ve: CD14, CD19, CD34, CD45, CD133, HLA-DR	-	High-density micromass cultures, followed by safronin O staining	NA	↑	-	[42]
MesenCult™-XF, Stem Cell Technologies	hBMSC, hADSC	+ve: CD73, CD90, CD105, HLA-I −ve: CD3, CD14, CD31, CD34, CD45, CD80	NA	NA	NA	-	-	[43]
	hBMSC, hADSC	+ve: CD73, CD105, CD90, HIL-A-I −ve: CD3, CD14, CD34, CD45, HIL-A-II, CD80	NA	NA	NA	↑	-	[44]
	hUCSC	+ve: CD73, CD90, CD105 −ve: CD14, CD34, CD45, CD79a, HLA-DR	-	High-density micromass cultures, followed by alcian blue staining	14 days or longer	-	-	[45]
	hBMSC	+ve: CD90, CD166	↑	3D aggregate chondrogenic induction, followed by aggrecan gene expression via RT-PCR	21 days	NA	NA	[46]
TeSR-E8 medium, StemCell Technologies	hADSC, hBMSC, hDPMSC	+ve: CD73, CD90, CD105, CD44	NA	NA	NA	↑	↓	[47]
RoosterNourish MSC-XF xeno-free media, Roosterbio	hBMSC	NA	NA	NA	NA	-	-	[48]
High Performance XF Media Kit, RoosterBio	hBMSC	+ve: CD73, CD90, CD105, CD166 −ve: CD14, CD34, CD45	-	3D aggregate chondrogenic induction, followed by alcian blue staining	NA	-	-	[49]
SF medium sc-82,013-G, Haoyang	hPDSC	+ve: CD73, CD90, CD105, CD44	NA	NA	NA	-	-	[50]
Human AB serum	hADSC	+ve: CD54, CD73, CD90, CD105 −ve: CF14, CD19, CD34, CD45, CD80 and HLA-DR	-	High-density micromass cultures, followed by alcian blue staining	15 days	-	-	[51]
SF SPE-IV defined medium, ABCell-Bio	hADSC, hBMSC, hUCSC, hDPMSC	+ve: CD10, CD13, CD29, CD44, CD73, CD90, CD105, CD166, D7-Fib, HLA-ABC, CD15, CD49a, CD56, CD106, CD146, CD340 −ve: CD14, CD31, CD33, Cd34, CD45, CD79a, CD133, CD184, HLA-DR, HLA-G, CD271, alpha10ITG, Stro-1, MSCA-1	-	3D aggregate chondrogenic induction, followed by immunohistochemical COLII staining	28 days	-	-	[52]
MSCGM-CD, Lonza	hPDSC	+ve: CD73, CD90, CD105 −ve: CD14, CD19, CD34, CD45, HLA-DR	-	3D aggregate chondrogenic induction, followed by histological alcian blue staining	21 days	-	-	[53]
Allogenic human umbilical cord serum	hESC	+ve: CD29, CD34, CD73, CD90, CD105 −ve: CD45	NA	NA	NA	-	-	[54]
Homemade chemically defined medium	hUCSC	+ve: CD13, CD29, CD44, CD73, CD90, CD105, CD166, HLA-ABC −ve: CD14, CD19, CD34, CD45, HLA-DR	NA	NA	NA	↑	-	[55]
XcytePLUS™ media	hWJMSC	+ve: CD73, CD90 and CD105 −ve: CD34, CD45	-	3D aggregate chondrogenic induction, followed by alcian blue staining	21 days	↑	↑	[56]
hPL	hADSC	NA	↑	2D induction culture followed by alcian blue staining	7, 14, and 21 days	↑	NA	[57]
	hADSC	+ve: CD29, CD73, CD90, CD105 −ve: CD34, CD45, CD31	↑	3D aggregate chondrogenic induction, followed by histological alcian blue staining and total RNA quantification for COLI, COLII, aggrecan and matrilin 1	14 days	↑	↓	[58]
	hBMSC	+ve: CD13, CD29, CD44, CD49e, CD73, CD90, CD105, HLA-ABC −ve:CD14, CD19, CD45, HLA-DR	-	3D aggregate chondrogenic induction, followed by alcian blue histological staining	14 days	-	-	[59]
	Human MSC, Lonza	+ve: CD73, CD90, CD105 −ve: CD34, CD45, HLA-DR	-	High-density micromass cultures, followed by alcian blue staining	21 days	↑	-	[60]
	hADSC	+ve: CD29, CDD44, CD73, CD90, CD105, CD166, HLA-I −ve: CD31, HLA-DR	↑	High-density micromass cultures, followed by alcian blue staining	47 h	-	-	[61]
Knockout™ Serum Replacement, Invitrogen	hBMSC	+ve: CD29, CD44, CD73, CD90	-	3D aggregate chondrogenic induction, followed by aggrecan histological staining and immunohistochemical staining for COLI, COLII and COLX	28 days	-	-	[62]
Allogeneic human serum	hWJMSC	+ve: CD44, CD73, CD90, CD105 −ve: CD34, HLA-DR	-	2D culture followed by alcian blue staining	18–20 days	-	-	[63]

**Table 2 ijms-25-10627-t002:** MSC characterization surface marker at P5.

Medium	Donor	CD44		CD90		CD105		CD146	
		**%Positive**	**Intensity**	**%Positive**	**Intensity**	**%Positive**	**Intensity**	**%Positive**	**Intensity**
FBS	Donor 1	100.00%	426,771	99.89%	136783	99.27%	485,392	99.08%	47,395
	Donor 2	100.00%	515,385	99.98%	55726	98.99%	548,620	99.56%	97,516
	Donor 3	98.45%	356,204	99.96%	111981	99.13%	540,016	99.58%	75,128
ACF	Donor 1	100.00%	386,393	99.97%	28899	99.97%	106,591	97.04%	94,525
	Donor 2	100.00%	341,008	100.00%	32065	99.98%	92,335	96.88%	73,116
	Donor 3	100.00%	306,765	99.98%	24265	99.99%	123,255	99.96%	108,784
XF	Donor 1	99.92%	435,363	99.91%	110798	99.98%	227,249	68.30%	21,207
	Donor 2	99.91%	404,182	99.97%	91289	99.96%	94,691	89.73%	35,109
	Donor 3	99.97%	440,183	100.00%	121705	99.81%	114,696	70.50%	29,508

**Table 3 ijms-25-10627-t003:** The donor summary.

	Age	Gender	Race	Company
Donor 1	19	F	C	Lonza
Donor 2	24	M	B	Lonza
Donor 3	37	M	C	Stem Cell Technologies

**Table 4 ijms-25-10627-t004:** Primers used for qPCR.

Primer	Forward Sequences	Reverse Sequences
GAPDH	ACAACTTTGGTATCGTGGAAGG	GCCATCACGCCACAGTTTC
P16	GGGTTTTCGTGGTTCACATCC	CTAGACGCTGGCTCCTCAGTA
P21	AGTCAGTTCCTTGTGGAGCC	GCATGGGTTCTGACGGACAT
P53	ACAGCTTTGAGGTGCGTGTTT	CCCTTTCTTGCGGAGATTCTCT

## Data Availability

The raw data supporting the conclusions of this article will be made available by the authors on request.

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
