# Peer review of "Comparative Analysis of Serum and Serum-Free Medium Cultured Mesenchymal Stromal Cells for Cartilage Repair"

_ijms, 2024, doi:10.3390/ijms251910627_

Round 1

Reviewer 1 Report

Comments and Suggestions for Authors

In this work, the Authors compared the in vivo regeneration and in vitro ability to differentiate of MSC culture in presence (FBS-MSC) or in absence (SF-MSC) of serum using, for the in vivo experiments, a rat osteochondral defect model. Their results suggest that MSC expanded in serum free medium presented variations in morphology, surface markers, senescence status, and differentiation capacity; then the users need to perform thorough investigation on chondrogenic potency before to employ expanded MSC for cartilage reconstruction. The experiments were well planned and comprehensive; the results clearly presented. The paper can be accepted for publication after these minor revisions.

 MINOR REVISIONS:

Line 51: burden[11] space missed

Lines 77,123,143,253: in vivo should be in italic

Figures 1 and 6: in optical microscopy pictures scale bar is missed. Furthermore, more histological details are needed (e.g. for figure 1 indicate where the collagen II and cartilage deposition are)

The Donors evaluate were 3. I wonder why in some pictures the Au reported the results of all the Donors, while in other the represented just 2 Donors (figures 1 and 2)

Figure 5: is it a representative picture? If yes the Au have to specify.

Author Response

Comment 1: Line 51: burden[11] space missed

Response: Edited.

Comment 2: Lines 77,123,143,253: in vivo should be in italic

Response: Text format amended.

Comment 3: Figures 1 and 6: in optical microscopy pictures scale bar is missed. Furthermore, more histological details are needed (e.g. for figure 1 indicate where the collagen II and cartilage deposition are)

Response: Missing scale bars added to the figures and respective scale bar dimensions added to the figure caption. For histological details, additional description added to Figure 1 caption (Line 126-127): Hyaline cartilage was stained red in Safranin O staining and brown color in Collagen II immunobiological staining. Scale bar = 1 mm.

Comment 4: The Donors evaluated were 3. I wonder why in some pictures the Au reported the results of all the Donors, while in other the represented just 2 Donors (figures 1 and 2)

Response: For in vivo study (Figure 1 and 2), we have validated a consistent poor cartilage regeneration using ACF medium-expanded MSC with two Donors (Donor 1 and Donor 2), which we think is sufficient to support the conclusion that ACF medium is not suitable for MSC expansion destined for cartilage repair therapies.

For in vitro study, we used three donors and three media to conduct a comprehensive experiment. This gives us more comparative results on the necessity for the in vitro chondrogenic potency evaluation for serum-free medium from different commercial brands.

Comment 5: Figure 5: is it a representative picture? If yes the Au have to specify.

Response: Yes, Figure 5 shows representative figure on Donor 2, similar pattern was also found in the other two donors, specification has been added to the main text (Line 203-204): These morphological changes were consistent across all three donors. Figure 5 shows their representative morphology with Donor 2.

Reviewer 2 Report

Comments and Suggestions for Authors

This manuscript compared the in vitro and in vivo behaviors of MSC cultured with serum-free medium or serum medium. The author said that the regeneration of damaged cartilage is effective in cells cultured with the serum medium. However, it was said that the proliferation and differentiation efficacy of cells in vitro is more effective in serum-free medium.

After culturing the cells, the author transplanted them into animals without differentiation. However, in order to compare the efficacy of the serum medium and the serum-free medium in this study, the others experimental groups needed. MSC have to grown with a serum medium or a SF medium, and then induced chondrogenic differentiation with a chondrogenic differentiation medium, then transplanted.

In addition, in the results of this study, the mechanisms showing differences in in vitro differentiation and in vivo treatment efficacy were not mentioned at all.

And it is necessary to explain whether cells cultured in serum medium and cells cultured in serum-free medium show differences in therapeutic effects.

Author Response

Comment 1: This manuscript compared the in vitro and in vivo behaviors of MSC cultured with serum-free medium or serum medium. The author said that the regeneration of damaged cartilage is effective in cells cultured with the serum medium. However, it was said that the proliferation and differentiation efficacy of cells in vitro is more effective in serum-free medium.

Response: Please note that we have employed two different serum-free medium in this study: MesenCult™-ACF Plus Medium (Stem Cell Technologies, Vancouver, Canada), which we refer in our manuscript as ACF medium, and Custom MSC SF XF Medium Kit (ME20236L1, Gibco), which we refer as XF medium. For in vivo study, we found that ACF medium expanded MSC showed inferior cartilage regeneration compared to MSCs derived from conventional 10% FBS DMEM. For in vitro study, we validated that ACF medium, although supported higher proliferation of MSCs, resulted in MSC with poor chondrogenic ability (Figure 6A), relative to XF- and FBS-MSCs, correlates well to the in vivo cartilage repair outcomes.

The main focuses of our study were to demonstrate that, although the serum-free medium derived MSCs showed better proliferation properties in vitro (which was demonstrated by both ACF-MSC and XF-MSC), this does not automatically correlate to their in vivo cartilage therapeutic effect (as demonstrated by the poor repair efficacy of ACF-MSC). Thus, prior in vitro validation for specific therapeutic applications is necessary for all clinical trials, when clinicians are considering an unfamiliar serum-free medium.

Comment 2: After culturing the cells, the author transplanted them into animals without differentiation. However, in order to compare the efficacy of the serum medium and the serum-free medium in this study, the others experimental groups needed. MSC has to grow with a serum medium or a SF medium, and then induced chondrogenic differentiation with a chondrogenic differentiation medium, then transplanted.

Response: Intra-articular injection undifferentiated MSC has been applied to many clinical trials. In our case, we selected the rat osteochondral defect model and the treatment method of implanting a hydrogel-MSC (undifferentiated) mix, which we have previously demonstrated to be able to regenerate cartilage [reference 24]. Although no prior chondrogenic differentiation induction was performed to the implanted MSCs, we believed that the experiment was well-controlled since the cell undifferentiated status and the delivery method were consistent for both FBS-MSCs and ACF-MSCs. Moreover, our results show that FBS-MSC was able to enhance cartilage repair even without the pre-differentiation step, but not with ACF-MSC. The despairing repair outcomes between FBS-MSC and ACF-MSC was consistent with MSC donors. The in vitro chondrogenesis induction further validates that MSCs cultured in ACF medium have inferior chondrogenic potencies. We believed that these results are coherent and strong enough to distinguish the efficacy of the specific serum-free medium for MSC expansion in cartilage regeneration.

Comment 3:  In addition, in the results of this study, the mechanisms showing differences in in vitro differentiation and in vivo treatment efficacy were not mentioned at all.

Response: Our in vitro differentiation result of ACF-MSCs and FBS-MSCs was coherent with their performance in in vivo study. Poor in vivo repair outcome of ACF-MSCs relative to FBS-MSCs was corroborated to the poor chondrogenic potential of ACF-MSCs relative to FBS-MSCs in vitro (Fig 6A, B). For in vitro studies, we included serum free media from another company (XF) to demonstrate that not all serum free media are the same. Some do, as in the case of XF, support chondrogenic potential of the expanded MSC (XF-MSC, Fig 6A, B). As for the drastic different outcomes between the two serum free media, although we are interested in exploring the mechanisms, further investigation is hindered by the lack of knowledge of the specific formulation of these serum-free media due to their proprietary property.

Comment 4: And it is necessary to explain whether cells cultured in serum medium and cells cultured in serum-free medium show differences in therapeutic effects.

Response: We showed difference in the surface markers expression (Table 2) and the senescence/apoptotic status (Fig 7) of MSC cultured in serum medium and serum-free medium. Speculation on how these differences could affect the differentiation potential of the expanded cells were included in the Discussion (line 269-301).

Response for improvements in Introduction: We think that the introduction section is provided with sufficient background knowledge from the general application of mesenchymal stromal cells (MSCs), the current shift in MSC manufacturing focus to serum-free culture and the specific gap in chondrogenesis potency validation of serum-free medium. Please clarify if any part of the Introduction section requires further elaboration or if any references need to be added.

Reviewer 3 Report

Comments and Suggestions for Authors

The manuscript titled "Comparative Analysis of Serum and Serum-Free Cultured Mesenchymal Stem Cells for Cartilage Repair" presents important findings on the potential of mesenchymal stem cells (MSCs) for cartilage repair. The study highlights the necessity of shifting from fetal bovine serum (FBS) to fully defined serum-free (SF) media to eliminate issues like xeno-contamination and batch-to-batch variability, which is crucial for clinical applications. However, there are a few areas that need improvement for clarity and accuracy:

1. Supplementary Material: The three digits on the left side appear unnecessary and should be removed.

2. Abstract: The phrase "we first compared" is redundant in an original research article and should be removed.

3. The criteria for defining MSCs need to be more rigorous. Please reference the position statement by the ISCT: Mesenchymal stem versus stromal cells: International Society for Cell & Gene Therapy (ISCT®) Mesenchymal Stromal Cell committee position statement on nomenclature 2019 by Viswanathan, S. et al.

4. The discussion should adopt a more critical tone regarding the outcomes of clinical trials. While many trials have been conducted, results are often unsatisfactory, and this should be reflected.

5. The term "donor" is misleading as it suggests that MSCs were isolated by the authors. Since the cells were purchased commercially, they should be referred to as "cell lines."

6. Similarly, on the graphs, replace "donor 1," "donor 2," etc., with the actual names of the cell lines to avoid confusion.

Figure 5:  Clarify what the arrows in Figure 5 are indicating.

Figure 3: Provide clear explanations for the abbreviations used in Figure 3.

Table 1: The section of the introduction that refers to Table 1, along with the table itself, would be better suited in the discussion section.

Comments on the Quality of English Language

 Minor editing of English language required.

Author Response

Comment 1: Supplementary Material: The three digits on the left side appear unnecessary and should be removed.

Response: Amended.

Comment 2:  Abstract: The phrase "we first compared" is redundant in an original research article and should be removed.

Response: Amended

Comment 3:  The criteria for defining MSCs need to be more rigorous. Please reference the position statement by the ISCT: Mesenchymal stem versus stromal cells: International Society for Cell & Gene Therapy (ISCT®) Mesenchymal Stromal Cell committee position statement on nomenclature 2019 by Viswanathan, S. et al.

Response: Thank you for pointing this out. We have amended the definition of MSC from mesenchymal stem cell to mesenchymal stromal cell according to the statement by ISCT.

Comment 4. The discussion should adopt a more critical tone regarding the outcomes of clinical trials. While many trials have been conducted, results are often unsatisfactory, and this should be reflected.

Response: We have added a sentence in the concluding paragraph to stress this point (Line 335-336): which could be responsible for the unsatisfied outcome in a substantial number of clinical trials.

Comment 5&6. The term "donor" is misleading as it suggests that MSCs were isolated by the authors. Since the cells were purchased commercially, they should be referred to as "cell lines." Similarly, on the graphs, replace "donor 1," "donor 2," etc., with the actual names of the cell lines to avoid confusion.

Response: The MSCs were purchased from companies, however, they are non-genetically modified primary cells extracted from different donors, as listed in Table 3.  Since the term “cell lines” commonly refer to immortalized cells, we prefer to use ‘donor’ to represent primary MSCs.

Comment 7: Figure 5:  Clarify what the arrows in Figure 5 are indicating.

Response: We have included arrows in Figure 5, and indicate in figure caption (Line 206): Black arrow indicates vacuolar structures.

Comment 8: Figure 3: Provide clear explanations for the abbreviations used in Figure 3.

Response: We added the abbreviation explanations in Figure 3 caption (Line 178-179): ✦: small star-shaped MSC (RS); ▲: long-fibroblast-like spindle-shaped MSC (SS); ■: flattened cuboid-shaped MSC (FC).

Comment 9: Table 1: The section of the introduction that refers to Table 1, along with the table itself, would be better suited in the discussion section.

Response: We would like to keep Table 1 in the Introduction as the table provides the background and rationale for this study.

Response for improvements in Introduction: We think that the introduction section is provided with sufficient background knowledge from the general application of mesenchymal stromal cells (MSCs), the current shift in MSC manufacturing focus to serum-free culture and the specific gap in chondrogenesis potency validation of serum-free medium. Please clarify if any part of the Introduction section requires further elaboration or if any references need to be added.

Response for minor editing of English language required: Our members have further checked on the English language of this manuscript and highlight the emendation in the text.

Round 2

Reviewer 2 Report

Comments and Suggestions for Authors

This manuscirpt is a research article, not a review. Table 1 summarizes various media and various culture technologies(monolayer and 3D culture), but, the direct association with this manuscrip is low.

The author explains that MSCs which was cultured with serum medium have improved healing efficacy than it cultured with serum-free medium. If so, the results of predicting and analyzing the cause should be presented.

For example, it is necessary to analyze differences between serum and serum free cultured cells through gemomics or proteomics analysis, and to analyze major adhesion molecule or integrin change through a method of knock out or inhibitors for the genes.

Author Response

Comment: This manuscirpt is a research article, not a review. Table 1 summarizes various media and various culture technologies(monolayer and 3D culture), but, the direct association with this manuscrip is low. 

Response: One of the aims of our study is to evaluate the performance of commercial serum-free (SF) medium for in vivo cartilage repair. As outlined in our Introduction (Lines 81-90), there is a significant gap in the literature regarding the chondrogenic potential of SF media. We found no comprehensive reviews on this topic, which is why Table 1 was created to support our study’s objective.

Table 1 summarizes the various SF media and chondrogenesis induction culture techniques (monolayer and 3D). This highlights that, despite different chondrogenesis induction methods, the evaluation of SF media often lacks robustness—many studies fail to include FBS-MSC control groups or rely on superficial assays which are neither quantitative nor qualitative. Additionally, a number of studies neglects chondrogenesis performance altogether, omitting differentiation assays. Therefore, we believe the table is relevant to the research objectives presented in our manuscript and complements the Introduction. We also believe this table serves as a valuable summary of most MSC SF medium studies since 2011 and will be helpful to fellow clinicians and researchers in the field.

However, we acknowledge that this information could potentially burden the main text. As such, we are open to moving the table to the supplementary materials while maintaining it as important background content. We would like to consult the editorial board on this matter.

Comment: The author explains that MSCs which was cultured with serum medium have improved healing efficacy than it cultured with serum-free medium. If so, the results of predicting and analyzing the cause should be presented.

For example, it is necessary to analyze differences between serum and serum free cultured cells through gemomics or proteomics analysis, and to analyze major adhesion molecule or integrin change through a method of knock out or inhibitors for the genes.

Response: As presented in our first revision, we identified several plausible pathways related to MSC cartilage repair performance, such as senescence, apoptosis status, and surface marker expression. In our study, FBS-MSCs served as the baseline control group, following the conventional MSC culture protocol. Our results demonstrated that MesenCult™-ACF Plus Medium (Stem Cell Technologies, Vancouver, Canada), a large-scale commercial SF medium, produced poor cartilage repair outcomes, which aligns with its lower in vitro chondrogenic potency. We also observed that chondrogenic potency varies across different SF media, as evidenced by the superior performance of XF-MSCs expanded in Custom MSC SF XF Medium Kit (ME20236L1, Gibco).

We did not conduct genomic or proteomic analyses to investigate the cellular pathways responsible for the poor performance of ACF-MSCs or the high performance of XF-MSCs. This is primarily because extensive research has already explored cellular pathways and chondrogenic potency, as discussed in our manuscript. Additionally, the proprietary nature of the unique formulations in these SF media, which contain undisclosed growth factors and bioactive compounds, limits the feasibility of analyzing their specific effects on MSC cellular-genomic pathways. Studying the combinatory effects of these compounds would likely not yield precise or novel insights into MSC chondrogenesis. Instead, we focused on the functional performance of MSCs derived from these media, which provides more practical and actionable information.

While we agree that investigating the underlying mechanisms could enrich the study (we included a couple of statements in the discussion, Line 334-338), this suggestion falls outside the scope and objectives of our current research. Furthermore, due to funding and resource limitations, we are unable to pursue these experiments at this time. Nonetheless, we believe that our findings offer valuable insights that can guide clinicians and researchers in selecting SF media for MSC applications.

Round 3

Reviewer 2 Report

Comments and Suggestions for Authors

I confirmed the author's sufficient correction and response. In future studies, it is recommended to perform an analysis of important mechanisms.